# Silicone-Thioxanthone: A Multifunctionalized Visible Light Photoinitiator with an Ability to Modify the Cured Polymers

**DOI:** 10.3390/polym11040695

**Published:** 2019-04-16

**Authors:** Qingqing Wu, Wen Liao, Ying Xiong, Jianjing Yang, Zhen Li, Hongding Tang

**Affiliations:** Engineering Research Center of Organosilicon Compounds & Materials, Ministry of Education, College of Chemistry and Molecular Sciences, Wuhan University, Wuhan 430072, China; 2011202030085@whu.edu.cn (Q.W.); 2015202030090@whu.edu.cn (W.L.); yingxiong@whu.edu.cn (Y.X.); 2013202030068@whu.edu.cn (J.Y.); lizhen@whu.edu.cn (Z.L.)

**Keywords:** photopolymerization, visible light photoinitiator, silicone, thioxanthone, modification, thermal stability, resistance against water/acid

## Abstract

A silicone-thioxanthone (STX) visible light photoinitiator was prepared by the nucleophilic substitution reaction of 2-[(4-hydroxybenzyl)-(methyl)-amino]-9H-thioxanthen-9-one (TX-HB) and γ-chloropropylmethylpolysiloxane-*co*-dimethyl-polysiloxane (PSO-Cl). Its structure was confirmed by ^1^H NMR, ^13^C NMR, FTIR, UV-vis and GPC. The photopolymerization kinetics of 1, 6-Hexanedioldiacrylate (HDDA) and trimethylolpropane triacrylate (TMPTA) initiated by STX confirmed that STX is an efficient photoinitiator. Its visible light photolysis experiment and the photopolymerization kinetics studies implied that a possible synergistic effect existed between two adjacent thioxanthone groups. Moreover, a higher migration stability was revealed in STX than 2-benzyl (methyl) amino-9H-thioxanthen-9-one (TX-B). STX could change the surface property of the cured film of polyurethane diacrylate prepolymer (PUA) from hydrophilic to hydrophobic, as well as change the thermal stability of the polymer network. Meanwhile, it could improve the resistance against water and acid. Thus, STX is an effective multifunctionalized photoinitiator.

## 1. Introduction

In recent years, photopolymerization has been widely recognized for its many applications in coatings, printing ink, adhesives, 3D print, and optoelectronic products [1,2,3]. As an important component in the polymerization process, the photoinitiator or the photoinitiator system plays an important role in the polymerization process. Among these photoinitiators, polymeric [4], polymerizable [5,6,7,8,9,10,11,12] and one-component photoinitiators [13,14,15] have garnered much more attention in comparison to the conventional low molecular weight photoinitiators due to their low odor, non-yellowing, nontoxicity, low migration, and good compatibility with the formulation components [16,17].

In general, a photoinitiator is the producer of active species after light irradiation and then initiates polymerization. If the photoinitiator has an ability to play another role in the material after photopolymerization finishes, it should be great material design from the point of view of green chemistry. Thus, various specific functional groups may be incorporated into the photoinitiator which could be used as the multifunctionalized photoinitiator. And it is desirable for the future development of photopolymerization. Indeed, some photoinitiators with the ability to change surface properties of cured films have been reported [18,19]. It is still a challenge but attractive task to develop low odor, non-yellowing, nontoxicity, low migration, good compatibility with formulation components and multi-functional photoinitiators for photopolymerization.

Silicone has been known as an inorganic–organic hybrid polymer material featured with a series of properties such as good chemical and thermal ability, weather-resistance, low friction coefficient, low surface energy, and nonstick behavior [20,21,22,23,24]. It has also been widely used as an additive agent to improve the properties of polymer materials, such as thermal ability [25,26,27] and weather-resistance [28,29]. As a kind of multifunctional photoinitiator, silicone-containing photoinitiator could also endow some specific properties on the photocured materials, such as reducing the critical surface tension of the crosslinked films [30,31]. Up to now, almost all the reported silicone-containing photoinitiators are sensitive to the UV light [31]. Recent researches in photopolymerization technique have paid much more attention to the visible light system since visible light is cheap, safe and possesses high penetration ability in the presence of ultraviolet absorbing monomers, pigments, and substrates. Thus, it is important and necessary to explore the visible light silicone-containing photoinitiators for future applications. Recently, we reported on a series of silicone-naphthalimide visible photoinitiator for free radical polymerization. They not only showed very good photoinitiating properties, but also brought about the desired modification to the cured-materials, such as improved resistance against water and ethanol, and better thermal stability [9,10].

Thioxanthones (TXs) are the important visible photoinitiator candidates [32,33,34,35,36]. We have reported a series of one-component aminothioxanthone (ATX) visible light photoinitiators, such as small molecular (TX-A, TX-B, TX-C and TX-Ac in [Fig polymers-11-00695-ch001]) [37], waterborne (TX-MPEG in [Fig polymers-11-00695-ch001]) [38], and polymerizable (TX-PA, TX-EA and TX-BDA in [Fig polymers-11-00695-ch001]) [36,37,38,39,40,41] TXs. They showed good visible light photoinitiating properties in the absence of additional hydrogen donors. As a part of our continuing interest in the development of thioxanthone (ATX)-based visible photoinitiating systems for radical photopolymerization, the integrated photoinitiator system of silicone and thioxanthone as a multifunctionalized photointiator STX (Scheme 1) was designed and prepared, in which ATX was introduced in the polysiloxane side chains (Scheme 2). Its photophysical properties, photopolymerization behavior, as well as impacts on the properties of polymer films have been investigated. Its photophysical data indicates that the obvious molecular synergistic effect exists between the two adjacent TX moieties anchored on the silicone backbone. The investigation into its photopolymerization behavior confirms that it is an efficient visible photoinitiator with comparable initiating ability to small molecular TX-B and commercial used CQ/MDEA. Moreover, the surface properties of the cured film of PUA films changes from hydrophilic to hydrophobic after the photocuring process. Meanwhile, this multifunctionalized photoinitiator STX improves the thermal stability of the cured polyurethane diacrylate (PUA) films and effectively increases the water- and acid-resistance abilities.

## 2. Experimental

### 2.1. Materials

2-[(4-Hydroxybenzyl)(methyl)amino]-9H-thioxanthen-9-one (TX-HB) [39] and 2-benzyl(methyl)amino-9H-thioxanthen-9-one (TX-B) [37] were prepared following the literature methods. Polyurethane diacrylate (PUA) prepolymer was kindly offered free by Jiangmen Ever Ray Environmental Material Co. Ltd, Jiangmen, China. *N*, *N*-Dimethylformamide (DMF) was dried and purified according to the standard laboratory methods. 1, 6-Hexanedioldiacrylate (HDDA, 90%), trimethylolpropane triacrylate (TMPTA, 85%) was obtained from Aladdin. Camphorquinone (CQ, 98%) was obtained from Heowns. All other reagents and solvents were obtained from Aladdin and used as received.

### 2.2. Instrumentation

^1^H (300 MHz) and ^13^C (75 MHz) NMR spectra were determined at room temperature on a VARIAN Mercury 300 spectrometer of the Spectropole (Varian, Salt Lake City, UT, USA). The molecular weight of STX was determined by gel permeation chromatography (GPC) in THF solution using a Waters 2960D Separation Module Containing Waters 515 pump, HR1, HR3 and HR4 THF columns and a Waters 2414 Refractive Index Detector (Waters, Milford, CT, USA) with a calibration curve for polystyrene standards. The properties of UV-absorption were measured by UV-visible spectrophotometer (Shimadzu UV-3600 spectrometer). The light source was assembled from xenon lamp (Laite optics, XD 300, cold light source or NBeT Group Corp., HSX-F300, Beijing, China) with a filter (λ > 400 nm). The light intensity was determined using a SRC-1000-TC-QZ-N reference monocrystalline silicon cell system (Oriel, Newport Corporation, Beijing, China), which was calibrated by National Renewable Energy Laboratory, A2LA accreditation certificate 2236.01. The contact angles of water on the polymer films were measured on a contact angle microscope (Kruss DSA100, Kruss, Humburg, Germany). Thermo gravimetric analysis (TGA) was conducted using Setsys 16 (Aetaram, France) from 30 to 600 °C at 10 °C/min heating rate in nitrogen atmosphere. Differential canning calorimeter (DSC) was performed on a DSC822e (Mettler Toledo, Switzerland) from 50 to 100 °C at a heating rate of 10 °C/min under nitrogen.

### 2.3. Synthesis

#### 2.3.1. Synthesis of γ-Chloropropylmethylpolysiloxane-*co*-dimethylpolysiloxane (PSO-Cl)

In a three-necked flask, a solution of γ-chloropropylmethyldimethoxysilane (7.31 g, 40 mmol), dimethoxydimethylsilane (28.85 g, 240 mmol) and 80 mL of THF was heated to 50 °C. Then a mixture of water (10.08 g, 560 mmol), 60 mL of THF and 0.10 g hydrochloric acid (37.5 %) was slowly drooped into the flask with stirring. The solution was stirred at 50 °C for 24 h. After that, the volatile substances were stripped off. Then, hexamethyldisiloxane (3.24 g, 20 mmol) and cation exchange resin (1.0 g, 5 wt%) were added into the system. The reaction mixture was stirred at 95 °C for 5 h. After the mixture was cooled down to room temperature, the cation exchange resin was removed by filtration. Stripping off the volatile substances gave 16.5 g of PSO-Cl; the yield was 65%. ^1^H NMR (CDCl_3_, 300 MHz) δ ppm: 0.05 (s, 6 H, -Si(CH_3_)_3_), 0.07 (s, 39 H, Si-Me), 0.61 (t, 2H, Si-CH_2_), 1.81 (m, 2H, -CH_2_-), and 3.50 (t, 2H, -CH_2_Cl). ^13^C NMR (CDCl_3_, 75 MHz) δ ppm: 47.6 (-CH_2_Cl), 26.8 (-CH_2_-), 15.0 (Si-CH_2_), 1.8 (CH_2_-Si-**C**H_3_), 1.1 (Si-CH_3_), and −0.5 (-Si(CH_3_)_3_).

#### 2.3.2. Synthesis of Silicone-Thioxanthone (STX)

An oven-dried flask equipped with a condenser was charged under argon with PSO-Cl (3.18 g, 5 mmol Si-CH_2_CH_2_CH_2_Cl), TX-HB (2.08 g, 6 mmol), K_2_CO_3_ (0.83 g, 6 mmol), and 0.1 g NaI in 50 mL of DMF. The reaction mixture was stirred at 100 °C for 24 h. After the mixture was cooled down to room temperature, the inorganic salt was removed by filtration. Then, the volatile substances were stripped off under vacuum. The viscous mixture was purified by SiO_2_ column chromatography using ethyl acetate and petroleum ether (1:10 *v*/*v*) as an elution. The yellow sheet solid (1.80 g) was obtained with a yield of 38 %. ^1^H NMR (CDCl_3_, 300 MHz) δ ppm: 0.08–0.09 (s, 43H, Si-CH_3_), 0.65 (t, 2H, Si-CH_2_), 1.84 (m, 2H, -CH_2_-), 3.11 (s, 3H, -NCH_3_), 3.89 (t, 2H, -CH_2_O), 4.56 (s, 2H, -NCH_2_), 6.85 (d, 2H, *J* = 8.1 Hz, Ar-H), 7.09–7.15 (m, 3H, Ar-H), 7.36–7.42 (m, 2H, Ar-H), 7.54 (s, 2H, Ar-H), 7.94 (s, 1H, Ar-H), and 8.63 (d, 1H, J = 8.1 Hz, Ar-H). ^13^C NMR (CDCl_3_, 75 MHz) δ ppm: 179.9 (C=O), 158.2 (ArC), 148.3 (ArC), 137.7 (ArC), 131.5 (ArCH), 129.8 (ArC), 128.6 (ArCH), 127.9 (ArCH), 126.8 (ArCH), 125.9 (ArCH), 125.5 (ArC), 124.0 (ArCH), 118.9 (ArCH), 114.6 (ArCH), 110.4 (ArCH), 70.1 (-CH_2_O-), 55.7 (-NCH_2_), 38.5 (-NCH_3_), 22.8 (-CH_2_-), 13.0 (Si-CH_2_), 1.0 (Si-CH_3_), 0.9 (SiCH_3_), 0.7 (SiCH_3_), and −0.8 (Si(CH_3_)_3_). FT-IR (film, cm^−1^): ν = 1590 (C=O), 1262 (Si-CH_3_), 980–1150 (Si-O-Si), and 802 (Si-CH_3_). M_n_ = 2920 g/mol, M_w_ = 2930 g/mol, and Ð = 1.00. 

### 2.4. Visible Light Photolysis Experiments

The visible light photolysis of photoinitiators was carried out through the analysis of the changes in the absorption of the maximum wavelength in the visible region. The absorption spectra of the photoinitiators in THF were measured with a UV-visible spectrometer (Agilent 8453) under xenon lamp exposure (XD 300, I = 57 mW cm^−2^) at room temperature.

### 2.5. Photopolymerization Experiments

TMPTA and HDDA were used as low viscosity monomers. *N*-Methyldiethanolamine (MDEA) was used as a hydrogen donor if needed. The film polymerization experiments were carried out in laminated conditions. The photosensitive formulations were deposited on a KBr pellet in laminate for irradiation with xenon lamp (XD 300, I = 28 mW cm^−2^). Monomers, STX and hydrogen donors if necessary in the presence and absence of THF acted as a photopolymerization system. The evolution of double-bond content was continuously monitored by real-time FT-IR spectroscopy (Nicolet IS 10) at 1610–1650 cm^−1^. The double-bond conversions (DC) were calculated from Equation (1) [41]
(1)$$\mathrm{Conversion}\;\left(\%\right)=\left(1-\frac{A_{X,t}}{A_{X,0}}\cdot \frac{A_{ST,0}}{A_{ST,t}}\right)\times 100$$
where *A*_0_ and *A_t_* represent the area of the IR absorption peak of the functional group of the sample before and after exposure during time t. The subscripts ST and X represent the internal reference ester carbonyl peak at 1730 cm^−1^ and the double bond peak at 1610–1650 cm^−1^. For each sample, the series real-time FT-IR runs are repeated at least three times.

### 2.6. Migration/Extractability Study of STX in Polymer Films

HDDA polymer samples for migration study were prepared by a photopolymerization procedure under argon. The samples were made from HDDA by using the same amounts of photoinitiator ([PI] = 3 × 10^−5^ mol/g) and irradiating with visible light (HSX-F300, I = 200 mW cm^−2^) for 40 min. After that, the samples were grounded into small particles and then immersed in 10 mL of acetone for 3 days at room temperature. The solids were filtered off and the solution was taken to measure the leached photoinitiators by fluorescence spectroscopy. Fluorescence intensities (F) for TX-B and TX-PSO were measured at 540 nm with exciting at 450 nm. Calibration curves were built by plotting the changes of fluorescence intensity F versus TX-B and TX-PSO concentrations.

### 2.7. Preparation of Polyurethane Films Initiated by TX-B and STX

Photoinitiator and PUA were dissolved in acetone thoroughly. The solvent was removed under vacuum and the mixture was dispensed on a pre-cleaned glass slid and allowed to spread to obtain a liquid resin film of desired thickness. Then it was exposed to visible light for 0.5 h to obtain the polymer film.

### 2.8. Water and Acid Resistance of Cured PUA Films

Water and acid resistance of cured PUA films were evaluated. The weighed cured films (*W*_0_) were immersed in the distilled water or 50 % H_2_SO_4_ (aq) at room temperature for 24 h, followed by wiping off the surface solvents with a piece of filter paper and tested for weight *W*_1_. The weight change percentage *A* was used to evaluate the water and acid resistance of the films.
(2)$$A=\frac{\left(W_{1}-W_{0}\right)}{W_{0}}\times 100\%$$

## 3. Results and Discussion

### 3.1. Synthesis and Characterization of STX

Gama-chloropropylmethylpolysiloxane-*co*-dimethylpolysiloxane (PSO-Cl) was obtained by the equilibrium reaction. In order to make it easier to introduce ATX moiety to polysiloxane, dimethylsilyloxy was used to dilute the chloropropylmethylsilyloxy. The ratio between dimethylsilyloxy and chloropropylmethylsilyloxy in PSO-Cl (Scheme 2) is about 6 (m = 1, n = 6, p = 3) according to the careful calculation of their integral areas in ^1^H NMR spectrum. Thus, the average molecular weight of PSO-Cl was easily calculated to be 1903.5 g/mol. Subsequently, STX was prepared by a convenient nucleophilic substitution reaction of TX-HB and PSO-Cl (Scheme 2). The structure of STX was confirmed by ^1^H NMR, ^13^C NMR, FTIR, and GPC. The comparative spectra of the ^1^H NMR and ^13^C NMR among TX-HB, PSO-Cl and STX are presented in the Appendix A. In comparison to the ^1^H NMR spectrum of STX to PSO-Cl, the triplet peak at 3.50 ppm for the methylene group in the chloropropyl of PSO-Cl disappears completely and is accompanied with a new triplet peak which appears at 3.89 ppm. For their ^13^C NMR spectra, the peak at 47.6 ppm for -CH_2_Cl in PSO-Cl disappears and the peak at 70.1 ppm appears. FTIR spectra of TX-B and STX are shown in Appendix A. In the FTIR spectrum of STX, the peak at 1590 cm^−1^ could be ascribed to the characteristic vibration of the carbonyl group in the thioxanthone moiety. The peaks at 1262 cm^−1^ and 802 cm^−1^ result from the characteristic vibration of Si-CH_3_ and the broad peaks at 980–1150 cm^−1^ could be ascribed to the characteristic peaks of Si-O-Si; these peaks belong to the polysiloxane segment of STX. All of these demonstrate that the chloropropyl in PSO-Cl has been completely substituted by phenolic hydroxyl and ATX has been bonded to the polysiloxane as the side chain.

According to the values of the integral area in ^1^H NMR spectrum of STX, the ratio of the dimethylsilyloxy group to thioxanthone group is about 6. It is consistent with the ratio of the dimethylsilyloxy group to chloropropylmethylsilyloxy group in PSO-Cl. Based on the average molecular weight of PSO-Cl, the average molecular weight of STX is speculated to be 2835 g/mol. The number average molecular weight of TX-PSO was measured to be 2920 g/mol by GPC. Therefore, ^1^H NMR and GPC data gave a consistent result. The average number of the TX moiety in STX is calculated to be about 3.0. The content of the TX moiety in STX was further confirmed by UV-Vis spectrum (Appendix A).

STX shows good solubility in common organic solvents, such as THF, dichloromethane, and ethyl acetate, implying that it would have excellent compatibility with other components of the photopolymerization system. STX exhibits very low solubility in water. The solubility of STX in water is measured to be only 1.71 × 10^−7^ mol/L = 4.99 × 10^−4^ g/L, which means 1 L deionized water dissolves 4.99 × 10^−4^ g STX at most.

### 3.2. Visible Light Photolysis

The visible light photolysis curves of STX in THF are shown in Figure 1a. The intensity of the absorption tends to decrease against the increase of visible light irradiation time. In comparison to the photolysis of TX-B (Figure 1b), some difference have been noted in the photolysis of STX. Firstly, the photolysis of STX is more evident during the first 0.5 h than that in TX-B. Secondly, the photolysis in TX-B is smooth relatively in the whole 2 h photolysis procedure after the first 0.5 h exposure. However, STX exhibits a rapid photolysis at the following 0.5 h accompanying with a new peak at 384 nm. After this rapid photolysis stage, After this rapid stage, the photolysis of STX slows over time. The possible procedure of photolysis of STX is proposed in Scheme 3. Upon the exposure of visible light, the photoinitiator (PI) absorbs protons to produce an excited singlet state (PI^1^) which could pass into an excited triplet state (PI^3^) through intersystem crossing. The excited triplet state molecule diffuses in close vicinity and works on the tertiary amine group of another thioxanthone moiety, which forms an encounter pair followed by the formation of a contact pair intermediate and the electron transfers to form a contact radical-ion pair. Back electron transfer could turn radical-ion pair back to the ground state of the photoinitiator. The proton transfer in contact with the radical-ion pair could produce the aminoalkyl radical and thioxanthyl ketyl radical. The possible exit procedure of the free radicals in the absence of polymerizable monomers is presented in Scheme 4. As can been seen from Scheme 4, the ketyl radicals undergo disproportionate or coupling reacting with themselves or aminoalkyl radicals. Accordingly, the conjugated system is damaged and photolysis happens. Meanwhile, the radicals could be oxidized by oxygen to form peroxide-free radicals (B(-H)OO). After that, they could react with hydrogen donors B to form peroxides (B(-H)OOH) and new free radicals. In PI/THF system, THF could be worked as the hydrogen donors. Therefore, much oxygen was consumed to oxidize THF. Since the back electron transfer process is thermodynamically favorable and is clearly faster than the proton transfer process especially in the presence of dissolved oxygen [39,40], only little aminoalkyl radical and thioxanthyl ketyl radical are produced. Therefore, the photolysis is little at the first 0.5 h relative to the whole photolysis procedure. After the dissolved oxygen is gradually consumed, the proton transfer process becomes faster and faster. In the STX system, the TX moieties have been anchored on the polysiloxane backbone. The soft main chain structure of polysiloxane and the limited movement enhance the interaction between the two adjacent TX moieties. Thus, the contact pair forms easier in STX than in TX-B through this synergistic effect ([Fig polymers-11-00695-ch002]). Therefore, the photolysis is more evident during the first 1 h. The new peak at 384 nm could be attributed to the absorption of the ketyl radical [39,40] since it is produced too fast to be transferred. After 1 h, the remaining TX moiety is about 60% in STX. Meanwhile, it is more and more difficult to interact to form a contact pair since the synergistic partners have been transferred. Therefore, after a rapid photolysis, it changes slower and slower. In the TX-B system, however, the remaining TX moieties are easier to find than their free partners, so the photolysis continues smoothly.

### 3.3. Photopolymerization

STX was used as the photoinitiator for the polymerization of TMPTA in laminate under the xenon lamp exposure (I = 28 mW cm^−2^). In order to decrease the viscosity of the TMPTA photopolymerization system, THF was used as the diluent. The photopolymerization kinetic profiles of TMPTA initiated by STX in different concentrations without hydrogen donor are shown in Figure 2a. As seen from Figure 2a, the increase of the initiator content makes a short induction period. It could be that the higher concentration of the photoinitiator could yield more radicals by the incident light, which is consistent with the photopolymerization behavior of TX-B (Appendix A). The higher concentration of free radicals could accelerate the photopolymerization rate sometimes, but it could also accelerate the recombination of primary radicals. So, the polymerization rate increases more and more slowly with the increase of the photoinitiator concentration. The gelation restricts the diffusion and mobility of macroradicals and pendant double bonds and slows down the rate of radical termination. This results in a buildup of radical species, promoting the rate of polymerization, leading to autoacceleration. However, the increased crosslinking level eventually limited the monomer mobility; the propagation reaction then also becomes diffusion-controlled along with radical termination. Thus, the overall polymerization rate decreases.

In the Type II photopolymerization system, amine always plays dual roles in the polymerization. It not only acts as hydrogen donors through electron transfer and proton transfer, but also could react with oxygen, thereby reducing the retarding effect of oxygen on polymerization [42,43]. Therefore, the polymerization of TMPTA initiated by STX with different concentrations of MDEA was measured. The photopolymerization kinetic curves are shown in Figure 2b. The results indicate that the photopolymerization rate and final conversion are increased and the induction period is shortened with the increase of the MDEA concentration. It could be that the higher concentration of MDEA, the more active radicals can be produced during the irradiation.

The results shown in Figure 2b were obtained in the presence of solvent THF. In order to evaluate the impact of THF on polymerization, polymerization without a solvent was also carried out. The polymerization kinetic curves of TMPTA initiated by STX with THF as a diluent or not are shown in Figure 2c. In the absence of MDEA, TMPTA exhibits a steeper polymerization curve without THF as a diluent. It could be that solvent THF could increase the mobility of active species and lead to a gentle autoacceleration effect. By contrast, in the presence of both MDEA and THF, STX exhibits a higher polymerization rate and final conversion. It could be attributable that in the presence of MDEA, the active species are so many that the coupling of these primary radicals is significant without solvents. The addition of THF increases the mobility and weakens the coupling of primary radicals. Therefore, in the presence of both MDEA and THF, STX had a better photoinitiation activity.

1, 6-Hexanedioldiacrylate (HDDA) was used as a low viscosity monomer for further evaluating the photoinitiating ability of STX. STX has better solubility in HDDA than in TMPTA. It was unnecessary to add the solvent to the photopolymerization system. The photopolymerization kinetic curves of HDDA initiated by STX and TX-B are shown in Figure 2d. For comparison to the commercial available visible photoinitiator, the photopolymerization kinetic of HDDA initiated by CQ/MDEA was also investigated (Figure 2d). A slightly higher photopolymerization rate and final conversion are observed in the STX system than that in the TX-B system. It could be that the synergistic effect among the attached thioxanthone moieties on the polysiloxane backbone plays an important role as shown in [Fig polymers-11-00695-ch002]. On the other hand, it could be due to the free volume effect, which is always caused by volume shrinkage. The volume shrinkage occurs at very fast polymerization and results in an increase of free-volume formation which could increase the mobility of the residual double bond and lead to a slightly higher final conversion [8,44]. In comparison to the CQ/MDEA system, the photopolymerization rate was slower but the final conversion was almost the same after 6.8 min in the STX system. These results indicate that STX is an efficient one-component visible light photoinitiator, thereby reducing the harm to the materials and environment by amine coinitiators.

### 3.4. Migration/Extractability of STX in Polymer Films

In order to investigate the extractability of STX in photocured materials, HDDA polymer film was prepared, and the residual photoinitiator in the cured polymer film was extracted by acetone. TX-B was used for comparison. Fluorescence spectrum was used to determine the content of photoinitiators in the acetone solution. The correlation of the fluorescence intensity (at 540nm) with a concentration of TX-B and STX and the fluorescence intensity of their extraction are shown in Figure 3. In comparison to the initial photoinitiator quality, the mass fraction of the extracted photoinitiator is 8.1% for STX and 54.0% for TX-B. This big difference between them could be attributed to the differences in the molecular weight [9,10]. This indicates that STX could be used as an efficient one-component visible light photoinitiator with low extractability of the photoinitiator in the cured materials. The low solubility of STX in water as well as the low extractability and low toxicity of macromolecular photoinitiators implies that STX should be a potential visible light photoinitiator used in food packing or biomedical fields [45].

### 3.5. Impacts of STX to Properties of Cured PUA Films

Polyurethane acrylates (PUA) have been known as UV-curable coating resins with good mechanical properties and high solid content in comparison with other UV-curable resins [46,47]. However, some of their shortcomings such as poor weathering, poor water resistance and poor thermal stability limit their applications.

The contact angles of the polymer films of polyurethane diacrylate prepolymer initiated by TX-B and STX were investigated by a contact angle microscope. The contact angle of a liquid droplet on a surface is a direct reflection of the hydrophilicity/hydrophobicity of the surface. When the water contact angle is above 90°, it displays a hydrophobic property; otherwise, it is hydrophilic. The contact angle (WCA) (θ) of water on the cured polymer film is shown in Table 1 and the contact angles of polymer film initiated by TX-B and high concentration of STX are shown in Appendix A. The contact angle of the polymer film initiated by TX-B was measured to be 81.8°, implying that its surface is hydrophilic. However, the contact angles of all polymer films initiated by STX are above 90°, which demonstrates that their surfaces are hydrophobic. Moreover, a higher concentration of STX in the photopolymerization system could bring about a more hydrophobic polymer film evidenced by a larger water contact angle.

Water and acid resistance of the cured PUA films were evaluated. The results are shown in Table 1. From Table 1, it can be seen that the water and acid absorption ratios of the cured PUA films initiated by TX-B are up to 2.17% and 1.12%. When 6 wt% STX is used as the photoinitiator, the water and acid absorption ratios of the cured PUA films are decreased to 0.36% and 0.12%, respectively. These results may illustrate that the cured PUA films containing silicon can effectively improve the water and acid resistance. With the increase of silicon content, its water and acid resistance are enhanced. It may be due to the hydrophobicity of the cured PUA films initiated by STX. Therefore, this photoinitiator containing silicon could improve the water and acid resisitence of PUA films.

The thermal stability of the cured polyurethane films initiated by TX-B and STX was measured by thermal gravimetric analysis (TGA). TGA analyses are shown in Appendix A and the results are summarized in Table 1. For comparison, TGA curves of TX-B and STX were also presented in Appendix A. The degradation temperatures (T_5_, 5% weight lost temperature) are 273 °C for TX-B and 229 °C for STX. This difference could be due to the low molecular weight material in STX. However, the degradation temperatures (T_50_, 50 % weight lost temperature) are measured to be 326 °C for TX-B and 373 °C for STX. The relative high T_50_ for STX may be ascribed to the presence of high thermal stability of Si-O compounds in STX [48,49]. The degradation temperature, T_5_, of the cured polymer films initiated by STX systems are higher than 290 °C. However, this temperature is measured to be as low as 279 °C in the cured polymer film photoinitiated by TX-B. Thus, the introduction of a little bit of polysiloxane increases the thermal stability of polymers. However, more polysiloxane does not bring about higher degradation temperature. When the content of STX increases to be 6 × 10^−5^ mol/g, the T_5_ shows a little decrease. This trend is more distinct in T_50_ (50% weight lost temperatures). On one hand, polysiloxane results in higher thermal stability. On the other hand, more photoinitiator increases the degree of branching of the polymer and leads to a decrease of molecular weight between two neighboring branching points, which decreases the thermal stability. When the content of STX is relatively low, the presence of polysiloxane increases the thermal stability. With increasing the content of STX, the crosslinking acts as the main issue which decreases the thermal stability.

The thermal behaviors of the cured polyurethane films initiated by TX-B and STX were measured by differential scanning calorimeter (DSC) (Table 1 and Appendix A). The glass transition temperature (T_g_) of the cured polymer films initiated by TX-B is measured to be 48 °C, but it decreases to 43 °C in the case of STX. The decreased value in the glass transition temperature (T_g_) could be attributed to the introduction of Si-containing segments, which makes the polymer chain move more smoothly [50,51,52].

## 4. Conclusions

In summary, a multifunctionalized one-component thioxanthone-polysiloxane visible light photoinitiator (STX) has been designed and prepared. Results indicate that STX could be used as an efficient one-component visible photoinitiator to initiate polymerization of HDDA and TMPTA. It has good migration stability and high efficiency in comparison to its small molecular counterpart TX-B. The application of STX as the photoinitiator in curing polyurethane diacrylate makes the surface of the cured polymer films change from hydrophilic to hydrophobic and could improve water and acid resistance. Meanwhile, the thermal stability of the cured polymer films is adjustable by controlling the dosage of STX. Therefore, STX is demonstrated as a multifunctionalized photoinitiator with a high initiating activity and high performance in cured materials.

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
