# Peer review of "Silicone-Thioxanthone: A Multifunctionalized Visible Light Photoinitiator with an Ability to Modify the Cured Polymers"

_polymers, 2019, doi:10.3390/polym11040695_

Round 1

Reviewer 1 Report

This is an interesting report on the synthesis and characterization of a silicone-thioxanthone photoinitiator for visible light-mediated photopolymerization reactions. The manuscript is well-presented and the experiments were well-conducted, allowing to support the conclusions of the work. There are some minor issues that could be improved and further discussed in the manuscript. Please see my comments/questions bellow.

1. Minor issues:

- Line 46: “series of characters such as good chemical..”; by “characters” the authors mean “characteristics or properties”?

- Line 59: “such as the improved resistance against water and ethanol, improved thermal stability”; check the grammar

- Chart 1: missing legend

- Line 74: “TX-B was used for comparison. fluorescence spectrum”

- Table 1: “c” is missing

2. Did the authors test the solubility of STX is water and/or heterogeneous solvents?

3. Considering the role of oxygen in photolysis, the experiments were also conducted under inert atmosphere? Under this condition, the polymerization kinetics can be affected?

4. In general, the discussion/comparison of the performance of STX with other photoinitiators either commercially available or developed by other groups can be improved. Although experiments were conducted with other PI’s as a control, authors can put the discussion in a broader perspective to emphasize/compare the main features of their PI. This can be discussed regarding several aspects such as the solubility, photolysis, cost, etc.

5. Did the authors determine the upper limit of STX concentration regarding the induction period versus curing depth? What is the maximum curing depth using a fixed concentration of STX? An experiment showing how STX concentration affects the curing depth of polymer films could be very useful in the manuscript.

6. In migration/extractability study, the amount of extracted STX is considerable lower than for TX-B. This means that STX is participating in the polymer network and/or being entrapped in a more extent than TX-B or the observed results are mainly attributed to the differences in the molecular weight? This is unclear in the manuscript.

7. How the authors envisage the application of STX in food packing or biomedical fields? Can comment on applications involving water-soluble and organic solvent systems?

8. Contact angle studies comparing STX and TX-B were performed using the same photoinitiator concentration? This is unclear.

9. STX could be used to improve the water and acidic resistance of crosslinked polymers, which is useful for certain applications. Did the authors test the influence of STX on the UV-light resistance of the crosslinked polymer? 

10. The increase in the thermal stability of STX compared to TX-B is statistically different? This should be commented on the discussion.

Author Response

The manuscript has been revised according to referee’s comments. The answers for the referee’ comments and the corresponding changes in the revised manuscript are presented as the attached file.

Reviewer 2 Report

Reviewer Comments for polymers- 475719” Silicone-thioxanthone: A multifunctionalized visible light photoinitiator with an ability to modify the cured polymers”

The authors present a novel multifunctional photoinitiator (PI), which cannot only cure under visible light but is also able to modify the resulting polymer properties. They have investigated the kinetics of the photopolymerization under different conditions and compared to small molecules PI´s. They have also investigated in how far the polymer properties may change by using their multifunctional PI and found that hydrophobicity and with this water and acid absorption was changed due to the introduction of silicone moieties.

I have just a few minor comments that should be addressed by the authors.

1) Abstract: Please introduce all abbreviations (HDDA and TMPTA are missing).

2) Introduction: Please give more information about CQ/MDEA since not all readers might be familiar with this PI. Full name and chemical structure of CQ would be helpful. In general, chemical structures for all used chemicals, including monomers, might be helpful for the reader.

3) Experimental, lines 88-89: TX-HB is mentioned twice. I believe that might be a mistake.

4) Synthesis: The yields for reactions would be appreciated.

5) Synthesis lines 188 and following: Here, the authors talk about NMR spectra and how integrals were used to confirm structure. I would like to see better NMR spectra in the supporting information. They are too small to see individual peaks and multiplets. In addition, the integrals should be shown here.

6) Results and Discussion, lines 60-62: The authors state that Fig 2d shows higher conversion and rate for STX, but the final conversion is about the same and the rates are just slightly differing from the TX-B. Please, rephrase.

7) Results and Discussion lines 75 and following: The text mentions figure 3 a and b, but the figure shows differently. Please, correct. In addition, the inset is too small and not readable. Please adjust this figure.

8) Results and Discussion, Migration/extractability: When looking at the figure, it appears that the intensity, and with this, the concentration of STX is much higher than the concentration of TX-B. Since the authors have used the same starting concentration for both PI´s, I am wondering why they report the mass fraction in the manuscript and not the concentration. That gives a wrong impression to the reader. Since the molar mass of STX is much higher, the mass fraction is higher, but this does not tell anything about the actual number of molecules leaching out. Please rephrase that paragraph.

9) Results and Discussion: The authors discuss in how far hydrophobicity and water /acid absorption change due to the use of STX. It would be interesting to see how other materials properties may change after the incorporation of silicone moieties. It would be especially interesting to see DSC and DMA curves for the same PI concentrations used for the other studies. 

Author Response

(The authors gave the same response as above.)

Reviewer 3 Report

Thermal properties of cured polyurethane films initiated by TX-B and STX have been presented in the insufficient way. The thermal properties of TX-B and STX should be presented. What’s more the authors should also clearly explain, if the temperature of  thermal decomposition of TX-B and STX is sufficient high to use investigated photo- initiators as a modifiers of common used cured polymers.   

Author Response

(The authors gave the same response as above.)
